# Bio-Inspired Muco-Adhesive Polymers for Drug Delivery Applications

**DOI:** 10.3390/polym14245459

**Published:** 2022-12-13

**Authors:** Zina Jawadi, Christine Yang, Ziyad S. Haidar, Peter L. Santa Maria, Solange Massa

**Affiliations:** 1Department of Otolaryngology—Head & Neck Surgery, Stanford University School of Medicine, Stanford, CA 94305, USA; 2BioMAT’X I+D+i (HAiDAR R&D&I LAB), Universidad de los Andes, Santiago 7620001, Chile; 3Centro de Investigación e Innovación Biomédica (CiiB), Universidad de los Andes, Santiago 7620001, Chile; 4Programa de Doctorado en BioMedicina, Facultad de Medicina, Universidad de los Andes, Santiago 7620001, Chile; 5Department of Biomaterials and BioEngineering, Facultad de Odontología, Universidad de los Andes, Santiago 7620001, Chile

**Keywords:** mucoadhesive polymers, mucoadhesion, drug delivery, bioinspiration, BioPolymers

## Abstract

Muco-adhesive drug delivery systems continue to be one of the most studied for controlled pharmacokinetics and pharmacodynamics. Briefly, muco-adhesive polymers, can be described as bio-polymers that adhere to the mucosal (mucus) surface layer, for an extended residency period of time at the site of application, by the help of interfacial forces resulting in improved drug delivery. When compared to traditional drug delivery systems, muco-adhesive carriers have the potential to enhance therapeutic performance and efficacy, locally and systematically, in oral, rectal, vaginal, amongst other routes. Yet, the achieving successful muco-adhesion in a novel polymeric drug delivery solution is a complex process involving key physico-chemico-mechanical parameters such as adsorption, wettability, polymer chain length, inter-penetration and cross-linking, to list a few. Hence, and in light of accruing progress, evidence and interest, during the last decade, this review aims to provide the reader with an overview of the theories, principles, properties, and underlying mechanisms of muco-adhesive polymers for pharmaceutics; *from basics to design to characterization to optimization to evaluation to market*. A special focus is devoted to recent advances incorporating bio-inspired polymers for designing controlled muco-adhesive drug delivery systems.

## 1. Introduction

The pursuit of the ideal drug delivery strategy has led to the exploration of a variety of platforms to facilitate highly desirable, controlled, efficient and personalized drug delivery [1,2,3]. Herein, biomimetics and bio-inspiration, or in other words, the way nature, with a minimum of resources, seems to design, assemble, and produce high-performance materials, is perhaps a valuable and interesting model and source of inspiration for the design of novel polymer-based and hybrid drug delivery systems with advanced application safety and therapeutic properties and functionalities [3]. Such might also help overcoming the limitations and avoiding the un-wanted side-effects of traditional platforms.

Bio-adhesion, or muco-adhesion, herein is defined as the adherence of materials, via attractive inter-facial forces, for an extended period of time with the mucosal (mucus) surface layer, typically from the oral route [1]. Originally pioneered by ophthalmologists to increase the bio-availability of drugs through controlled release drug delivery, muco-adhesive polymer technology has recently been used to enhance therapeutic residency and performance for systemic and local drug delivery systems through buccal, nasal, vaginal, and rectal routes [2,3,4,5,6]. Over the past decades, muco-adhesive polymer-based drug delivery has received a great deal of attention for devising novel pharmaceutical systems. Indeed, muco-adhesive drug delivery systems provide interesting therapeutic benefits. For example, they increase retention (residency or bio-availability) time at the mucosal site, by-passing the first-pass hepatic metabolism and modification of gastric enzymes; thereby allowing for greater control over the drug (or load) release rate and increased therapeutic effects of the dosage-forms [7]. Muco-adhesive (dosage forms) drug delivery systems also offer an extended capacity for controlled drug release due to the controlled localization of drug delivery into the site of adhesion/application [8]. It is perhaps noteworthy here in that mucosal sites have rapid absorption and high blood flow, leading to a higher perfusion rate through the blood vessels [9]. Furthermore, the intimate contact (bio-adhesion) at the residence site could facilitate one-way drug flux, thus incrementing barrier permeability, especially when compared to the skin (as an application site) [2]. Low enzymatic activity, as well, is important for drugs that tend to degrade easily or rapidly in acidic or alkaline environments, hence, such dosage form platforms further improve bio-availability of such pharmaceutical loads [10]. In addition, with small, flexible dosage forms that do not cause irritation, muco-adhesion has improved patient compliance (non-invasive and user-friendly tools) [11]. Muco-adhesive systems, which are erodible, tasteless, and accessible (low cost), can also reduce dosing frequency, thereby enhancing safety [12]. Finally, muco-adhesive systems can load or encapsulate challenging pharmacologic molecules, such as peptides, cytokines and oligonucleotides, which are high-molecular-weight sensitive [13]. Nonetheless, current barriers to successfully increasing bio-availability include permeation and enzymatic barriers of the mucosa itself as well as the turnover rate in the mucosa leading to the muco-adhesive system being washed away from the absorption site before complete unloading/release of the drug content [4]. therefore, controlling pharmaco-kinetics and -dynamics, is a critical challenge and an enduring R&D&I issue of current interest, for better muco-adhesive solutions.

Thus far, muco-adhesive systems have been investigated and dosage forms developed for a variety of therapeutic areas and applications, including diabetes, gingivitis (gum disease), xerostomia (or dry mouth), and aphthous ulceration [2]. Briefly, formulating an effective muco-adhesive system, depends on many factors, most importantly, the polymer(s) that has sufficient mucosal adhesive properties, biocompatibility, and adequate release rate mechanisms. Some novel polymers, such as gelatin, lectins, and thiols, demonstrate greater muco-adhesion and retention and do merit more investigation [8]. To date, the most common dosage designs include adhesive ointments, patches, films, tablets, and gels, mostly which are often targeted for buccal (intra-oral cheek surface) drug delivery [14]. Yet, muco-adhesive drug delivery systems have the potential for other or further applications. Indeed, muco-adhesive drug delivery systems have demonstrated advantages over traditional drug delivery systems, to date. Through careful consideration and manipulation of device design, muco-adhesive capacity and pharmaco-kinetics/-dynamics (including factors/parameters such as delivery route, dosed drug release, penetration, residency, bio-availability and therapeutic efficacy), can lead to or result in novel superior tool(s). Meaning, safe, efficacious, controlled, scalable, user-friendly, and cost-effective drug delivery system(s) with modulated and fine-tuned bio-physico-chemico-mechanical/rheological properties [4]. This comprehensive review focuses on bio-inspired muco-adhesive polymers and systems for such drug release or pharmaceutic delivery.

## 2. Muco-Adhesion Theories

Surface properties, such as wetting, swelling, water absorption rate, contact angle and surface energy, play a particularly important role in polymer-based muco-adhesive drug delivery systems. Design, formulation or fabrication, characterization, evaluation, fine-tuning/optimization, translation, and overall performance, are principal theories and concepts, revisited/updated in the subsequent section. Figure 1 below illustrates the *main* theories of muco-adhesion along the formats of muco-adhesives.

### 2.1. Wetting Theory

Wetting theory, briefly, suggests that adhesion results from the molecular contact between two materials and the surface forces that develop, henceforth explaining the behavior of liquid or low viscosity systems, specifically spreadability, which indicates the favorability of the liquid to “wet” the surface of the substrate, as a *predictive metric* of adhesion [15]. Accordingly, the theory proposes that the affinity of the attachment between formulation and the mucosal surface is due to the inter-molecular interactions and the surface tension (forces) between the liquid and the substrate surface [15]. Herein, spreadability is derived from the equilibrium between the adhesive (*liquid to surface*) and the cohesive (*liquid to liquid*) forces acting on the system, as seen by the liquid-to-solid contact angle [15]. hence, the adhesive forces between the liquid and the surface of the substrate enable the liquid to spread—“wetting” the surface, therefore, the cohesive forces within the liquid enable the liquid to repel contact with the surface, thereby maintaining the shape or form of the droplet [15,16]. Using Young’s equation, which measures the various surface tensions and contact angles when a system is at equilibrium, an adhesive bond can be predicted as a measure of the adhesion of the whole system [2,17], as follows:$$\gamma_{TA}=\gamma_{PT}+{\;\gamma }_{PA}\cos\left(\theta \right)$$
where γ refers to the free energy, *T* to the tissue substrate surface, *P* to the polymer/liquid, *A* to the vapor, and *θ* to the contact angle between the solid and liquid interface [2,18].

Moreover, the tendency for a single drop of liquid to spread over a substrate surface is inversely related to the contact angle [19]. Importantly, wetting is favorable with contact angles less than 90° and is unfavorable with contact angles greater than 90°, resulting in increased cohesive forces among a drop of liquid [19]. Thus, the strength of a muco-adhesive system would be attributable to the inter-molecular forces governing the spread of the formulation, system, or device over the mucosal surface.

### 2.2. Mechanical Inter-Locking Theory

The low surface tension and high spreading coefficient of some polymers allows them to rapidly conform to surface irregularities. Mechanical inter-locking theory, or mechanical keying (or mechanical non-jamming), assumes the role of chain inter-locking in achieving muco-adhesion. It proposes that the adhesion between a liquid and a rough or porous surface is due to mechanical inter-locking and increased surface rugosity, possibly contributing to the inherent strength of the adhesive interface by enhancing the plastic dissipation of energy at the interface [20]. By filling irregularities on a rough or porous substrate surface, the interfacial area increases significantly, creating a “composite” interfacial region [20,21]. This composite region has an intermediate strength between the adhesive and the oxide, possibly contributing to the toughness, strength, and stability of the interface [20,21]. Studies have shown that the substrate must be pre-treated for the proper topography to ensure suitable mechanical inter-locking by linking the substrate-to-surface topography with adhesion strength and energy expended in fracture [20,21]. However, this theory is not applicable with great extractability, such as in adhesive substrates with smooth surfaces, and whether it is the singular cause of greater adhesion, continue to be up for debate and is the subject of sustained scientific study [20,21]. Ultimately, the measured increase in joint strength may be attributable to improved inter-facial contact, enhanced energy dissipation (“charged” interaction between the mucoadhesive polymers, set at a certain pH) in the adhesive, or due to other unknown subsidiary causes [13] and may be assessed/evaluated, as follows:Optimum Joint Strength=Constant×Mechanical Interlocking Component×Interfacial Chemical Component

### 2.3. Adsorption Theory

Adsorption, which is the most widely applicable theory, proposes that sufficient molecular contact at the interface causes adhesion because of the inter-atomic and inter-molecular forces, such as primary and secondary bonds that form energy barriers and semi-permanent interactions between the substrate surfaces [22]. Herein, stronger *primary* bonds, such as ionic and covalent bonds, tend to form un-desirable permanent interactions with mucosal layers [22]. In contrast, weaker *secondary* bonds, such as hydrogen bonds, van der Waals forces, hydrophobic interactions, and electrostatic attraction (layer-by-layer self-assembly technique, for example), are, therefore, much more desirable [23,24,25]. The most common secondary forces are van der Waals forces, which are individually weak, yet when present in large quantities, will create stronger adhesion [25,26]. Chemical bonds may also form across the interface, termed “chemisorption”, in which adhesion is due to strong covalent bonding (ionic, covalent, or metallic bonding) across the interface [13]. Thus, to simplify, after an initial contact between two surfaces, the net result of either primary or secondary forces, is what results in *adhesion*.

### 2.4. Electronic Transfer Theory

Based on the electronic differences between the structures, the electronic transfer theory proposes that muco-adhesion is the consequence of the transfer of electrons between the substrate and the mucus membrane (which is negatively charged due to the structure of mucin), where attraction occurs between the glycoprotein mucin network at the mucosal site and the biomaterial or polymer [27]. The muco-adhesive polymer and the mucosal layer have a significant difference in electronic properties and structure, leading to a double layer of electrical charges at the interface of the mucosal layer and the polymer [27]. Thus, muco-adhesive strength depends on the engaging forces within the electronic double layer (between the positively-charged polymer and the negatively-charged mucus) due to the difference in the electronic properties, as opposed to high joint strength or secondary chemical interactions, proposed by the other theories [21,27]. In the investigated systems, adhesive force increases with the particle(s) diameter, and is based on the derived Coulomb’s inverse-square law, or simply Coulomb’s law of dependence (first published in 1785 by French physicist Charles-Augustin de Coulomb), quantifying the amount of electrostatic force (hence why also referred to as Coulomb force) between two stationary, electrically charged particles [26]. Simply, the law states that the magnitude of the electrostatic force (along the straight line joining the two charges) of attraction (if charges have different sign) or repulsion (if charges have the same sign) between two-point charges is directly proportional to the product of the magnitudes of charges and inversely proportional to the square of the distance between them. Further, the Coulomb *potential* (part of the closely-associated Quantum field theory) considers continuum states thereby describing the electron-proton scattering, as well as discrete bound states, implying the hydrogen atom, within a designed *dosage form* system.

### 2.5. Fracture Theory

On the contrary to the above-mentioned theories, the force involved in the “separation” of two adhered surfaces; i.e., the adhesive strength, or fracture theory, is dependent on the detachment force that is needed to separate the mucosal surface and the muco-adhesive system after adhesion [28], assuming the failure of the adhesion phenomenon at some given interface [13]. In other words, it assumes that the separation or fracture occurs at the interface between the muco-adhesive and mucus surfaces, theoretically. Adhesive failure occurs at the weakest component (considered as a cohesive failure) is the most used metric in studies determining mechanical quantification of muco-adhesion [26].

Herein, adhesive strength, or the detachment force, is denoted as (*s_m_*) and is calculated as the maximal force needed for detachment (*F_m_*) over the total surface area encompassed by the muco-adhesive system (*A*_0_) [26,29], as follows:$$s_{m}=F_{m}/A_{0}$$
where, the formula for the fracture strength (*σ*) can be calculated by Young’s modulus for elasticity (*E*), the fracture energy (*ε*), and the critical crack length (*L*), the peak of free energy [13,29], as follows:$$\sigma =\sqrt{\frac{E*\varepsilon }{L}}$$
where, Young’s modulus for elasticity (*E*) describes tensile strength calculated by the ratio of tensile strength to tensile strain. The elastic modulus of the system is calculated through Hooke’s Law by the stress (s) and the shear (e). Stress is calculated as force (*F*) over the area (*A*_0_), and shear is calculated as the change in system thickness (Δ*l*) over original thickness (*l*_0_) [29], as follows:$$E={\left[\frac{\sigma }{\varepsilon }\right]}_{\varepsilon \to 0}={\left[\frac{\frac{F}{A_{0}}}{\frac{\Delta l}{l_{0}}}\right]}_{\Delta l\to 0}$$
where, Fracture energy *ε* comes from the reversible adhesion work (*W_r_*) that is needed to create new fractured surfaces and from the irreversible adhesion work (*W_i_*) that is required for deformation until the fracture of the adhesion [29,30], as follows:$$\varepsilon =W_{r}+W_{i}$$

Generally, there are three *key* factors influencing muco-adhesive performance, and include the bio-polymeric factor, the environmental factor, and the physiological factor. Herein, while this theory is powerful for mechanical systems that require quantification (determining the mechanical strength of a particular muco-adhesive material), the system must be of a uniform material, with set and measurable physical dimensions [8,29]. Furthermore, the system must be measured at the point of failure of adhesion at a mucosal interface [31]. However, fractures more frequently occur near the surface at the weakest point, more likely the mucus layer or the hydrated mucosal region, yet rarely at the interface of the mucosal site itself [7]. Thus, this mechanism theory is rendered most useful for single component, rigid muco-adhesive materials, with a defined fracture at the interface, and without *any* penetration [7].

### 2.6. Diffusion Theory

The diffusion theory describes the inter-diffusion or inter-penetration between the muco-adhesive polymer chains and the mucin (mucus) glycoprotein chains, with a sufficient depth to create a semi-permanent adhesive bond. Accordingly, a semi-permanent adhesive bond is formed by the deep inter-penetration and entanglement of polymer and mucin chains (and the driving force for such a process is the concentration gradient), approximately 0.2–0.5 µm, and the adhesive force is proportional to the degree of penetration (dependent upon the time of contact and diffusion co-efficients of both interacting substances; inter-penetration) [13,17]. The penetration rate, herein, is affected by diffusion co-efficient, flexibility, muco-adhesive chain character, molecular weight, cross-linking density, mobility, contact time, expansion capacity and depth of inter-penetration. It can be stated that a better mutual solubility between the muco-adhesive and mucus results in stronger muco-adhesive bond [13,17].

Table 1 Summary of the Major Muco-Adhesion Theories—summarizes the main theories, principles of muco-adhesion, and the forces applicable to muco-adhesive systems, such as wetting, mechanical inter-locking/keying, adsorption, electronic, fracture as well as diffusion-interpenetration.

## 3. Mechanisms of Muco-Adhesion

In pharmaceutical sciences, muco-adhesion, as mentioned earlier, is the state in which a material and mucus or a mucous membrane are held together for extended period of time by inter-facial forces. Therefore, muco-adhesive drug delivery system can promote high local drug concentration by retaining an intimate contact at the absorption site, thereby, increasing the permeability of drugs including peptides and proteins. Briefly, muco-adhesion, a complex process, between the mucous membrane and the muco-adhesive material(s)/polymer(s) of choice, has two principal stages that affect modulating the effectiveness of the dosage form: (a) *contact* stage and (b) *consolidation* stage [7]. Figure 2 illustrates the 2-stage mechanism and factors involved in muco-adhesion and drug delivery.

In the *contact* stage, the muco-adhesive polymer films come in contact with the mucosal membrane, whereby the film dehydrates the mucosal gel layer nevertheless hydrates itself [7,32]. In this initial step to form muco-adhesion, the formulation spreads and swells, which enables deep and intimate contact with the mucosal membrane/layer [15,33]. Whereas the delivery system physically or mechanically attaches over the formulation in oral, ocular or vaginal sites, deposition is stimulated by aero-dynamics (inertial impaction process) in the membrane/anatomical site, such as with the nasal route (bronchi of the respiratory tract), with difficulties within the gastro-intestinal tract (i.e., the rectum) [33].

In the *consolidation* stage, moisture activates the muco-adhesive materials to “plasticize” the system, break free, and bond via van der Waals and hydrogen bonds [7,29,32]. The muco-adhesive strength is influenced by the polymer and by hydration of the polymer dosage form when in contact with the mucus gel layer [32]. The consolidation stage is explained by both, either the dehydration and/or the diffusion theories [29,32]. When it comes to the diffusion theory, the muco-adhesive polymer and glycoproteins of the mucus layer are in contact via secondary bond formation and chain inter-penetration [7,34]. Certain muco-adhesive properties are favorable for chemical and mechanical interactions, including hydrogen bond groups, high molecular weight, anionic surface charge, and surface-active properties [7]. According to the dehydration theory, when muco-adhesive polymers capable of gelation are placed into contact with an aqueous environment, water moves until equilibrium is established between the two layers due to osmotic pressure differences [29,32]. In other words, water motion, not macro-molecular chain inter-penetration, results in adhesive bond formation [29]. The strength of the bond is determined by the extent of inter-mixing to reach equilibrium [32]. The dehydration theory cannot be applied for formulations that are solid or extremely hydrated [29]. Overall, consolidation is a very important process to combat the adhesive failure of a formulation/fabricated system to eventually achieve a stronger and prolonged muco-adhesion effect. It is perhaps noteworthy to mention that chemical bonds can be formed between the mucosal layer and the muco-adhesive material [35]. Di-sulfide bonds may exist between thiomers and mucosal layers rich in cysteine [35]. Although less strong than covalent bonds, other interactions may occur, including ionic (between cationic polymers and acidic mucosal layers—*ionic strength can also regulate the electrostatic interactions*), hydrogen (with hydrophilic functional groups), van der Waals (between induced or permanent dipole–dipole), and hydrophobic (with non-polar groups in aqueous solution) [35,36].

## 4. Development of Muco-Adhesives and Muco-Adhesive Drug Delivery Systems

### 4.1. Evaluating Muco-Adhesion

The process of muco-adhesion and the development of muco-adhesives using a wide variety of methods and techniques, has been widely studied. Such themes are influenced by experiment design, the applicability and properties of various bio-polymers, drug dosage forms, and instrumental variables. The three main tests of muco-adhesion are tensile, shear strength, and peel strength; in vitro or ex vivo tests for efficacy of muco-adhesion [2,36,37,38]. further, in vivo imaging is used to evaluate muco-adhesion and delivery efficacy [2,37,38]. However, there is no standard apparatus for testing muco-adhesion, which results in numerous tests for muco-adhesion and a lack of uniformity and cohesion between test methods, and thereby renders comparing the reported findings/results rather difficult [2,37,38].

Generally, tensile strength is the most commonly used method or test to assess muco-adhesion [1,2,16]. Usually, in this technique, force is applied perpendicularly to the polymer-substrate interface, creating tensile stress, which equals pressure over the contact area, to determine the amount of force required to separate the two [2]. Another common test is the wash-off test [38]. herein, the polymer is attached to a slide in a USP tablet disintegration apparatus, which replicates the mucosa and moves the polymer up and down until complete detachment of the polymer from the mucosa replicates [18,38].

Furthermore, shear stress is similar to the tensile test in that stress is uniformly distributed over the adhesive joint and is intended to measure the mechanical properties of the entire system [2]. Pressure is equally distributed over the whole system, and force is required to detach the whole system, although shear stress redirects the forces along the joint interface [2]. After all, the peel strength stress focuses the force at the edge of the joint and calculates the energy or force resisting peeling [2]. For most muco-adhesive systems, the peel strength is not relevant, though it is very useful for patch-based systems [2]. Finally, in vivo tests for muco-adhesion are often limited by ethical and feasibility constraints [2]. nonetheless, one common in vivo technique is to monitor gastro-intestinal transit and residence times of muco-adhesive polymers, or drug release (tracking/bio-distribution) from in situ, pre-implanted devices [38]. Using radioactive labeling on the delivery system and regular measurement of radioactivity in the target area, the muco-adhesive capacity of the delivery system can be hence, approximated [37,38].

### 4.2. Delivery Sites and Routes for Muco-Adhesive Bio-Polymer Drug Delivery Systems

To prolong drug action, muco-adhesive dosage forms need to be in close contact with the absorbing surface to increase residence time [38]. Muco-adhesive polymers are typically delivered in sites such as the eye conjunctiva, oral cavity, nasal cavity, vagina, and the gastro-intestinal (GI) tract, since the mucosa lines these areas of the body [37,38]. Each route has different benefits based on the properties of the dosage form at the site. While buccal and sub-lingual sites have more rapid onsets by by-passing first-pass metabolism, they can be inconvenient for the person due to taste and food intake [7]. Microvilli in the GI tract may enhance absorption but may result in first-pass metabolism and requires consideration of stability [7]. Rectal and vaginal routes have excellent local drug absorption but are difficult to administer [7]. Nasal and ophthalmic sites can drain *mucociliae* that can clear out the dosage form [7]. The following section explores those anatomical sites and delivery sites in some detail.

#### 4.2.1. Buccal/Oral Cavity (Intra-Oral)

Even though the buccal cavity has a smaller surface of around 50 cm [2], its ease of access renders it a preferred location for drug delivery by by-passing hepatic first-pass metabolism and local oral lesion treatments [7,27]. Both, buccal and sub-lingual routes can provide direct access to the systemic circulation via bypassing first-pass metabolism; however, due to its smooth, immobile surface, the buccal mucosa is less permeable than the sub-lingual mucosa [7,37,38]. Consequently, whereas sub-lingual delivery tends to release the drug quickly, the muco-adhesive formulation is more controlled in the buccal mucosa and, consequently, better for such drug delivery (better/higher bio-availability) [7]. To avoid the saliva washing away the drug, the delivery system typically includes a film impermeable to water [7]. Given that this region has non-keratinized epithelia with more permeable tissue relative to the skin, drugs that have short half-lives, poor permeability, need for a sustained release effect, sensitive enzymatic degradation, or poor solubility, would benefit from being delivered via the oral cavity [38]. Buccal and sub-lingual drug delivery systems come in tablet-, film- or spray-formats, amongst others.

#### 4.2.2. Ocular Cavity (Eye Conjunctiva)

Generally, while the conjunctiva, cornea, anterior chamber, and iris usually respond well to topical treatment, the eyelids tend to more frequently require systemic treatment. Goblet cells secrete mucin in the eye conjunctiva but not in the cornea; therefore, muco-adhesive polymers would attach well to the conjunctival but not the corneal mucus (static vs. dynamic barriers) [38]. Ophthalmic dosage forms improve with prolonged contact with eye tissues [38]. Yet, due to tearing formation (and tear dilution), and eyelid blinking, the medication or loaded/released drug can be rapidly removed from the ocular cavity, resulting in poorer bio-availability, that can be then reduced using ocular inserts or patches [27].

#### 4.2.3. Reproductive Lumen (Vaginal and Rectal)

Vaginal and rectal lumen delivery has been considered for topical and systemic treatment [27,38]. Systemic vaginal delivery bypasses hepatic first-pass metabolism and can lower dosing frequencies by remaining in the vagina for prolonged periods during the day and night [27,37,38]. The surface area of the vagina is enhanced by many epithelial folds and micro-ridges that cover the epithelial cell surface [38]. One challenge in this route is the possibility of migration within the vagina or rectal lumen that would affect the specific location of delivery [27]. Muco-adhesive polymers help minimize this migration, thereby, improving the therapeutic efficacy [27]. Yet, this route can be deemed subpar and inefficient due to the prompt transit of the drug-containing delivery system past the absorption site [27].

#### 4.2.4. Nasal Cavity

The nasal cavity is an excellent histologic route for drug delivery. Similar to the buccal cavity, the nasal cavity enables muco-adhesive formulation development, and has gained attention and interest to promote dosage form residence time as well as improving intimacy of contact with absorptive membranes of the biological system [7]. Indeed, the surface area of the nasal mucosal layer ranges from 150 to 200 cm [2] and has enhanced muco-ciliary activity with foreign particles [7,38]. As mentioned earlier, the mucus (and membrane) is composed of a glycoprotein, mucin, with negative charges due to the presence of sialic acid residues. In addition, the nasal cavity has dense vascular networks, permeable membranes, and an absorption capability, by-passing the first-pass hepatic metabolism; rendering the rate and extent of absorption comparable with intravenous (I.V.) drug administration [38].

#### 4.2.5. The GI Tract

Besides that, drug delivery across the mucosa bypasses the first-pass hepatic metabolism, herein, this route facilitates avoiding the degradation of *gastro-intestinal* enzymes. Hence, localizing a drug in parts of the GI propelled the development of muco-adhesive systems [27,38]. The main goal of oral use would be to significantly increase the residence time of the drug for a localized effect and to enable dosing once a day [38]. Modulating the delivery transit time has been investigated extensively in the GI [27]. Muco-adhesive drug delivery platforms and solutions, therefore, can provide patients with a rapid drug absorption and good bioavailability due to the extensive surface area and high blood flow.

## 5. Muco-Adhesive Bio-Polymers: *From Basics to Applications*

### 5.1. Characteristics of Bio-Polymers

Generally, bio-polymers are held together by primary bonds (covalent bonds) and secondary bonds (van der Waals and hydrogen bonds). Co-polymers are polymers composed of two or more different types of monomers. Three types of polymers can adhere to biological surfaces: (1) polymers using non-covalent and non-specific electrostatic interactions; (2) polymers with hydrophilic functional groups that enable hydrogen bonding with the substrate; and (3) polymers that bind to receptor sites on the mucus surface [2]. Further, polymers may also be classified as hydrophilic, hydrogels, or thermoplastic [36]. Hydrophilic polymers, such as chitosan, methylcellulose, and plant gums, swell when in contact with water and undergo complete dissolution [36]. Hydrogels, such as sodium alginate and guar gum, are typically cross-linked polymers and have limited swelling capabilities [36]. Thermoplastic polymers include polymers that generate carboxylic acid groups while degrading, such as polyanhydrides [36]. Some polymers are utilized in muco-adhesive formulations, including synthetic, biocompatible, biodegradable, polyorthoesters, polyphosphoesters, polyanhydrides, and polyphosphazenes [36]. Because physical and chemical interactions can affect the adhesion between the polymer and the mucosal substrate, several properties should be considered when determining the optimal polymer, particularly for a bioadhesive (muco-adhesive) drug delivery system [37]. For example, the polymer (and any degradation by-product) should be non-irritant, non-toxic, and non-absorbable (biodegradable) [2]. Ideally, the polymer can form a strong non-covalent bond with the mucosal cell surface and adhere quickly to moist tissue with some specificity to the site [2]. The polymer should not decompose while in storage or on the shelf using the specific drug dosage form [2]. The polymer should be easily incorporated into the drug with no hindrance once released [2]. last but not least, for the product to remain competitive, the cost of the polymer should not be too expensive [2].

Briefly, muco-adhesive bio-polymers play a role in facilitating the desirable mucosal interactions (the mucosal layer is made up of mucus which is secreted by the goblet cells or glandular columnar epithelial cells and is visco-elastic), hence, there are certain characteristics of polymers (whether natural, synthetic or composite/hybrid) that affect the resulting muco-adhesion. Designing and developing muco-adhesive polymers may be affected by different biochemical and structural polymer-to-mucosa interactions, such as hydrophilicity, molecular weight, charge ± and the pH. Indeed, whilst muco-adhesive bio-polymers are mainly water-soluble in nature, some can also be water-insoluble, for example [2,27,37]. A summary of the properties of muco-adhesive polymers follows.

### 5.2. Factors Affecting Muco-Adhesion

A muco-adhesive bio-polymer ideally has the capability to become sticky and has chain flexibility at the mucosal pH and ionic strength, small enough to favor inter-penetration yet considerable enough for entanglement, with optimal molecular weight, and relevant degree and rate of polymer swelling [39].

#### 5.2.1. Hydrophilicity

Bio-adhesive polymers contain hydrophilic functional groups that enable hydrogen bonding with the substrate, expand (swell) in water, and enhance exposure to anchor sites [2]. Swollen bio-polymers are efficient at maximizing the distance between the chains, resulting in chain flexibility and efficient substrate penetration [2].

#### 5.2.2. Molecular Weight

Briefly, bio-polymers with low molecular weight allow for better inter-penetration while if with high molecular weight would allow for entanglements [2]. thereby, increasing the molecular weight (of the chosen bio-polymer or bio-polymers) improves muco-adhesion, because the polymers diffuse, inter-penetrate, and/or interlock with the mucosa at the site of application [27]. Although the optimal molecular weight to maximize muco-adhesion depends and varies on/with each type of polymer and its inherent adhesive character alongside the nature of the target tissue (and route), bio-adhesive forces increase with the molecular weight (up to 100 kDa); above which there is no further benefit [2,27]. 

#### 5.2.3. Cross-Linking and Swelling Factor

Swelling, which is needed to induce mobility, plays a key role in the desired inter-penetration between the bio-polymer and the mucosal substrate, since it exposes the muco-adhesive sites to hydrogen bonding [15]. herein, bio-polymer concentration, ionic strength, and water concentration can influence swelling [15]. Although loose cross-linking is preferred for a higher degree of swelling, over-hydration can decrease adhesion and clear the muco-adhesive system from the mucosa [2,15]. Hence, determining the rate of swelling (at design and development stage) can be important for an effective muco-adhesive drug delivery system with an *optimized* time of clearance and rate of drug release [15]. Further, a higher initial force of application increases inter-penetration and muco-adhesive strength. Higher initial contact time between the muco-adhesive and the substrate increases swelling and inter-penetration of the polymer chains [15]. This *initial* applied strength and contact time correlates with muco-adhesion, because controlled contact improves regulation, inter-penetration, and degree of swelling, as follows:$$Cross-Linking\;Density\;\propto \frac{1}{Degree\;of\;Swelling}$$

Therefore, there is an indirect relationship between the cross-link density and the degree of swelling, also known as hydration [2,15]. Basically, a decrease in the cross-link density increases flexibility and the hydration rate [2]. Besides, the polymer surface area is directly proportional to muco-adhesion [2]. Furthermore, formulation of the adhesion promoters can enrich the muco-adhesion of cross-linked polymers [2]. The theories and approaches of the muco-adhesion process, discussed above, apply herein.

#### 5.2.4. pH at the Muco-Adhesive Bio-Polymer-Substrate Interface

The pH level at the interface of the bio-polymer and the substrate can impact the adhesion of the bio-adhesives with ionizable groups and polyanions such as the carboxylic acids [2,37]. pH can cause changes in dissociation and hydration based on the polymer pH and ± charge [27]. Local pH levels that are higher than the polymer pK will result in large ionization and lower than the pK in unionization [2].

#### 5.2.5. Concentration of the Active Bio-Polymer

An optimum concentration of active bio-polymer can be used to maximize liquid muco-adhesive formulations [2]. As polymer concentration increases, access to the polymer chain decreases, so adhesion decreases due to the limited available chain length for inter-penetration [27]. Generally, the ideal polymer concentration is 1–2.5 percent molecular mass [27]. However, for solid dosages such as tablets, polymer concentration and muco-adhesion strength are directly related [2]. In addition, spatial conformation impacts polymer concentration [27]. Furthermore, an increase in the polymer chain length results in a greater polymer flexibility, penetration, entanglement, and so, muco-adhesion strength [27].

#### 5.2.6. Drug/Excipient Concentration

Drug concentration can impact muco-adhesion. Indeed, a study exploring the effect of propranolol hydrochloride on Carbopol found that limiting water in a system do increase elasticity and adhesion [2]. Further, an increase in the toluidine blue O concentration in Gantrez, has increased muco-adhesion to the cheek due to the electro-static interactions between the cationic drug and the anionic co-polymer [2].

#### 5.2.7. Mucin Turnover Rate

Physiological factors, such as the rate of mucus turnover, can also impact muco-adhesion in systems [2,37]. This turnover rate can be influenced by the disease state and the presence of a bio-adhesive device [2,40]. The mucin clearance system will decrease the residence time on the mucosal layer [40]. Both, cell renewal and mucin turnover rates depend on the type and location of the mucosa [40].

## 6. Bio-Inspired Polymers and Application in Drug Delivery

Some muco-adhesive polymers are inspired by biological materials. Polymers such as chitosan have established research while others such as aggregate silk glue are in the initial stages of development. Notably, a bio-inspired polymer is *not* necessarily natural-origin. For example, Carbopol 934P is a synthetic bio-inspired muco-adhesive polymer. The next section examines bio-inspired polymers. Table 2 summarizes the recent advancements in bio-inspired muco-adhesive drug delivery systems; whilst Figure 3 depicts the bio-inspired/-mimetic polymers for use in muco-adhesive drug delivery.

### 6.1. Chitosan

Chitosan is a natural, biocompatible, and biodegradable polysaccharide, derived from shell-fish, that has been extensively used for wound closure, hemostatic, and other applications and that is soluble in aqueous solutions with pH less than 6.5 [41,42]. Lehr et al. evaluated the muco-adhesive properties of chitosan in vitro by measuring detachment forces for swollen pig intestinal mucosa polymer films in saline media [43]. Whereas natural polymers hydroxypropyl- and carboxymethylcellulose exhibited almost no muco-adhesion, the cationic polymer chitosan was muco-adhesive relative to Polycarbophil [43]. Further, He et al. evaluated the muco-adhesive properties of chitosan microspheres in vitro by measuring the mucin absorbed on microspheres and using turbidimetric measurements [44]. Adsorption studies between mucin and chitosan microspheres with varying crosslinking studies indicated strong interactions [44]. Chitosan microspheres were also retained in biological tissue [44]. In another study, Sogias et al. investigated the interactions between chitosan and gastric mucin to understand why this polymer is muco-adhesive [42]. Decreasing the number of amino groups enhanced the pH solubility window of chitosan yet decreased its ability to aggregate mucin [42]. Although the electrostatic attraction interactions between chitosan and gastric mucin can be inhibited with 0.2 M sodium chloride, these forces do not suppress aggregation of mucin particles when chitosan is present. [42] The impact of mucin aggregation in the presence of chitosan in solutions containing 8M urea or 10% *v*/*v* ethanol suggests that hydrogen bonding and hydrophobic effects, respectively impact muco-adhesion [42]. Previously, Xu et al. used chitosan to create adhesive hydrogels [45,46,47]. Bio-adhesives based on chitosan exhibited strong adhesive properties to mucosal tissues and minimal cytotoxicity [41]. Xu et al. found that mixing chitosan with catechol-containing compounds, 3,4-dihydroxy-L-phenylalanine (DOPA), hydrocaffeic acid (HCA), or dopamine has been shown to increase muco-adhesive properties and swelling [46]. The hydrogel factor was highest in the presence of DOPA and dopamine but lower in HCA [46]. The HCA-chitosan hydrogel had slow catechol release, perhaps due to electrostatic interactions between chitosan and HCA [46]. Additionally, a decrease in hydrogel swelling and HCA release increased muco-adhesion in HCA-chitosan hydrogen when compared to chitosan hydrogels only, in rabbit small intestine [46]. In addition, oxidation of HCA during contact with the mucosa enhanced muco-adhesion of HCA-chitosan hydrogels [46]. This approach is bio-inspired, because muco-adhesion of chitosan hydrogels in wet conditions was increased by adding catechol-containing compounds [46,47]. In addition, Kim et al. reacted chitosan with 3,4-dihydroxy hydrocinnamic acid-mediated and 1-ethyl-3-(3-dimethylaminopropyl)-carbodiimide hydrochloride to synthesize a chitosan-catechol conjugate [48]. This method was inspired by mussel adhesion to surfaces [48]. Mucin-particle interaction, turbidimetry, surface plasmon resonance spectroscopy, rheological characterizations, and in vivo fluorescence imaging techniques were used to measure the muco-adhesive properties in mice [48]. Compared to chitosan and polyacrylic acid separately, the chitosan-catechol conjugate yielded better muco-adhesion [48]. It is noteworthy that chitosan-catechol, typically, is prepared via chemical, electrochemical, or enzymatic syntheses [49]. The notable properties of chitosan-catechol include solubility, tissue adhesion, mechanical strength, and biocompatibility [49]. In addition, this conjugate can be prepared in different forms, such as hydrogels, films, sponges, microparticles, or even nanoparticles [49]. The development of mimics (biomimetics/biomimicry) such as polyethyleneimine–catechol, chitosan–catechol, and catecholic polymers was inspired by the abundance of catecholamine in mussel adhesive proteins [49]. For example, Yamada et al. synthesized chitosan-catechol adhesives that are resistant to water and inspired by insect cuticular sclerotization [49,50]. Rheologically, tyrosinase-catalyzed reactions and subsequent un-catalyzed reactions increased the viscosity of the chitosan solutions with adhesive shear strengths over 400 kPa [50]. Moreover, chitosan is limited in its poor load-bearing capability in hydrated conditions and dissolvability in pH conditions lower than 6 [51]. Zvarec et al. synthesized and prepared nanoparticle composites inspired by and mimicking the mechanical strength of squid beaks and mussel thread coatings using chitosan [51]. Finally, the inorganic γ-Fe_2_O_3_ and catechol organic interfaces enhanced the coordination between bonding and stability, at high temperatures and physiological pH conditions [51].

### 6.2. Mussel Adhesive Protein (MAP)

MAP, or mussel adhesive protein, is a 130-kDa protein that adheres to underwater surfaces [52]. MAP is likely muco-adhesive due to it containing DOPA, which has a hydrogen bond capability that has been attributed to its ability to interact with mucosal surfaces [53]. Indeed, MAP was shown to exhibit strong muco-adhesive properties in a thin film by Schnurrer et al. and in soluble form by Deacon et al. [52,53,54]. MAP is produced and inspired by blue mussel *Mytilus edulis* and perhaps worthy of further research.

### 6.3. Alginate-PEGAc

Previously, Davidovich-Pinhas et al. synthesized a muco-adhesive polymer labeled Alginate-PEGAc whereby the alginate backbone holds acrylate polyethylenglycol [55]. This polymer demonstrates the strength, simplicity, and gelation useful for muco-adhesion [55]. briefly, alginate (from algae) is biocompatible and bioactive with low toxicity [56]. The combination of PEG inter-penetration with the mucosal surface layer and the Michael-type addition reaction between the polymer acrylate end group and mucosal glycoprotein sulfide end group, strong bonding between the polymer and mucosal layer occurred/resulted [55]. The formation of this polymer was demonstrated through nuclear magnetic resonance, lack of cytotoxicity was tested through cell viability assays in vitro, and its muco-adhesive properties were evaluated through adhesion assays [55]. further, scanning electron microscopy was used to characterize the alginate-PEGAc; a synthetic polymer inspired by protein PEGylation, today, is used for controlled drug release and has since transformed the food and the biomedical industries [55,56].

### 6.4. Pectin-Sodium Carboxymethyl Cellulose System

Gupta et al. demonstrated the ability to use polypeptides to orally deliver salmon calcitonin using muco-adhesive polymers salmon calcitonin in vivo [57]. This system was inspired by the design of transdermal patches and was prepared by creating a matrix of carbopol, pectin, and sodium carboxymethylcellulose (1:1:2) and coating the matrix on three of four sides with impermeable, flexible ethyl cellulose backing layer [57]. The polymer matrix system had strong muco-adhesion to porcine small intestine in vitro, withstanding forces up to 100 times the overall weight of the system [57]. This system is used to deliver drugs orally, which was previously rendered challenging due to poor stability in the stomach and permeation across the intestine [57]. both, pectin and sodium carboxymethylcellulose are natural compounds, that are also to be held or classified as ‘*bio-inspired*’.

### 6.5. Carbopol 934P

Carbopol gels in water have also been evaluated for their muco-adhesive properties [58]. Tamburic et al. evaluated Carbopol 934, rheologically, with continuous shear, creep, and oscillatory measurements [59]. When compared to other gel systems, Carbopol 934 had the highest degree of gel network elasticity and viscosity, with low thixotropy [59]. Tamburic et al. also investigated the muco-adhesive properties of different polyacrylic acid gel systems [60]. When compared to EX-214 and Noveon AA-1, Carbopols 934P and 974P had the greatest muco-adhesive strength and had small differences between these two systems due to the neutralizing agents [60]. In addition, there was a correlation between muco-adhesive strength and rheological tan δ (phase lag) values [60]. Furthermore, Carbopol has been combined with other polymers to develop a drug delivery system with a mucoadhesive-controlled release [58]. Bera et al. utilized the mucoadhesive properties of the carboxypolymethylene (CP934P) polymer to synthesize glipizide microbeads regulating blood sugar in patients with diabetes [58]. Herein, CP934P is a biodegradable, biocompatible carbopol polymer that is useful for its swellability (swelling capability), biosafety, low cost, and lack of absorption by tissues [58]. Indeed, CP934P was able to reduce the fasting blood glucose levels in rats and guinea pigs and had high muco-adhesivity [58]. On the other hand, Blanco-Fuente et al. studied the bio-adhesive properties of carbopol hydrogels intended for buccal administration of propranolol HC1 in vitro with a tensile tester in/under different hydration conditions [61]. Briefly, the limited water in the system resulted in an increased adhesion [61]. Additionally, an increase in polymer molecular weight and crosslinking percentage decreased adhesion capacity [61]. When propranolol HC1 was added to the hydrogens, adhesion increased in the limited water system, since the polymer-drug complex formation enhanced elasticity [61]. When propranolol HC1 was added to the hydrogens, adhesion increased in the limited water system, since the polymer-drug complex formation enhanced elasticity, and decreased in the system with no water limitation due to precipitation of the carbopol-propranolol HC1 complex [61]. In another study, Takeuchi et al. evaluated the effectiveness of the muco-adhesive properties of carbopol-coated liposomes in orally administering calcitonin to rats [62]. Muco-adhesive liposomes coated with carbopol polymers and with chitosan polymers were prepared in rat intestines [62]. The adhesive property of liposomes coated with carbopol polymers was inversely related to pH levels of the dispersing medium, perhaps due to electric repulsion between the carbopol liposomes and the mucosal layer [62]. Administering carbopol-coated liposomes with calcitonin prolonged the reduction of calcium concentration in the blood [62]. Dodou et al. designed a synthetic muco-adhesive polymer that can move down the colon [63]. Safe propagation requires generating friction with the colonic surface [63]. Micro-patterning muco-adhesive films were able to generate high static friction in vitro and prevented damaging the colonic surface [63]. This was inspired by adhesives secreted by sea stars in the form of sponges released by tubular feet [63].

### 6.6. Spider Silk

As a strong polymer that is biocompatible, biodegradable, non-toxic, and lightweight, spider silk has great potential to be used as an adhesive material [64]. Spider silks have inspired several polymer blends, including 4RepCT variants, aggregate silk, and pyriform silk, yet require further research to explore the potential of each polymer as a mucoadhesive [64]. For example, Peng et al. adopted water-soluble recombinant spider dragline silk protein to create spinning dope [65]. The artificial spider silk was spun via a bio-inspired microfluidic chip mimicking natural spinning [65]. The tensile strength was 510 MPa, and elongation of the fibers was 15% [65]. Future advances have the potential to result in use.

### 6.7. Spider Silk 4RepCT Variants

Petrou et al. proved spider silk material as muco-adhesive by genetically engineering two variants of spider silk protein 4RepCT that exhibited functional muco-adhesive properties, including mucin binding properties and electrostatic interactions [64]. Petrou et al. speculate that these variants may be used individually or together with bio-functional silk proteins to build protein-based materials in mucosal treatments [64]. As a newly synthesized polymer inspired by spider silk, further research is required to characterize it, including the strength of the interactions and the ability to deliver drugs [64].

### 6.8. Aggregate Silk Glue

Visco-elastic aggregate silk glue from orb-weaver spiders has the potential to be used as muco-adhesive polymers due to their adhesive properties, for instance in high humidity conditions or situations [45]. The primary sequence of aggregate silk proteins lacks DOPA that strengthens adhesion [45]. Yet, the Dahlquist criteria for adhesives define robust adhesion to have Young’s modulus lower than 100 kPa; atomic force microscopy measurements suggest an average Young’s modulus of 70 ± 47 kPa for aggregate silk glue [45]. The strong adhesion and elasticity of this silk-based glue might lie in the structural hierarchy, a high percentage of charged amino acid, and motif structure enabling mobility needed for mucosal swelling and interaction [45]. The composition, structure, and environment-dependent behavior of aggregate silk could be useful as a “silk-based *bio-mimetic* glue”.

### 6.9. Pyriform Silk

Similar to aggregate silk, pyriform silk is a natural polymer that is inspired by spiders [66]. Although not specifically studied in the context of muco-adhesion, its properties render pyriform silk a potential muco-adhesive polymer. Spider threads have diverse mechanical properties that are useful biologically [66]. According to Wolff et al., pyriform spins into attachment discs, which are utilized to anchor silken threads to substrates, and is a biodegradable, biocompatible, and versatile polymer [67]. In addition, spiders produce silk as solid fiber with high polar content and fibrous cement components with charged amino acids similar to muco-adhesives [45]. For example, Blasingame et al. reported that the Pyriform Spidroin 1 of black widow spiders as a gene, enabled anchoring silk fibers in attachment discs to solid substrates [66]. Another gene—Pyriform Spidroin 2—was noted in orb-weaving attachment discs by Geurts et al. to spin synthetic spider silk fibers into a viscous liquid that rapidly solidifies and glues these fibers to substrates [68]. The cement part that spiders yield can serve as a visco-elastic fluid that can be used to provide the high contact area needed for hydrogen bonding and adhesive strength [45,67]. Indeed, pyriform is a versatile natural polymer that is biodegradable and biocompatible [67]. Wolff et al. found that the anisotropy and “hierarchical organization” of pyriform silk enabled the silk bio-adhesives to uniquely enhance adhesion strength [67]. In addition, the adhesion strength of the attachment discs can be modulated by spinneret movements macroscopically [67]. Pyriform polymerizes under ambient conditions that are functional in <1 s, and is stable for years [67].

### 6.10. Silkworm

Silkworms are more accessible to commercialize, because they can be formed, unlike spiders [45]. Silkworms produce a fiber with a fibroin core and sericin coating [45]. Silkworm fibroin was shown to be adhesive as a pH-sensitive hydrogel through electro-gelation, or e-gels [45]. The strength of the e-gels, inspired by silkworms, gives their potential to be mucoadhesive [45]. Serban et al. synthesized solubilized silkworm fibroin with polyethylene glycol, which had strong adhesive properties [69]. This new class of blend with silk fibroin and polyethylene glycol is cytocompatible, crosslinks within seconds, and can potentially stabilize through β-sheet silk formation [69]. Silk-polyethylene glycol sealants have comparable or better adhesive strength, lower swelling, and longer degradation times compared to polyethylene glycol sealants alone [69]. Kundu et al. designed a muco-adhesive film by combining silk fibroin with hydroxy propyl methyl cellulose and with poly ethylene glycol 400 for transmucosal drug delivery [70]. These fabricated stable muco-adhesive films were used as a vehicle for trans-mucosal delivery [70]. Increase in fibroin content of the films enhanced mechanical properties, ex vivo bio-adhesive strength, water stability, degree of swelling, and stability of films in simulated saliva needed for fast muco-adhesion [70]. Wei et al. bio-mimicked the silkworm spinning process by applying bio-inspired dry spinning equipment to spin regenerated silk fibroin fibers from aqueous solutions in air [71]. Spinning dopes with pH levels of 5.2 to 6.9 exhibited high spinnability [71]. Dry spinning experiments resulted in a breaking strength of 46 MPa under optimal conditions and could be increased to 359 MPa after being drawn into 80 vol.% ethanol aqueous solution [71]. In another study, Lou et al. fabricated bio-inspired microfluidic concentrators based on the photolithography process [72]. Silk glands and spinning ducts of silkworms were bio-mimicked to design microchips for regenerated silk fibroin aqueous solution [72]. The microfluidic channel and silkworm enriched the regenerated silk fibroin concentration [72]. Further, Kundu et al. prepared fibroin cryo-gels inspired by muga silkworm *Antheraea assamensis* using 2 fabrication temperatures for liver tissue engineering. The cryo-gels could support the viability of the human hepatocarcinoma cells in Live/Dead assays [73].

### 6.11. Sericin

Inspired by silkworm silk, sericin can be blended with polymers like sodium alginate and polyvinyl alcohol to contribute to drug delivery [74,75]. Although sericin is adhesive without the fibroin core produced in the silkworm fiber, conjugation of sericin with other polymers stabilizes the structure and minimizes immunogenicity [74,76]. Blending sericin with alginate has been used as a natural, biodegradable mucoadhesive polymer matrix for drug delivery [74,76]. Wang et al. explored sericin in the context of tissue engineering by fabricating and characterizing covalently crosslinked 3-D natural silk protein sericin hydrogel to deliver cells and drugs [77]. This hydrogel is injectable, enabling minimally invasive implantations, and is photoluminescent, hence, useful for bio-imaging and in vivo tracking [77]. Its cell-adhesiveness makes it effective for promoting cell attachment and proliferation [77]. The hydrogel has high porosity, elasticity, and pH-dependent dynamics which are useful in drug delivery [77]. Moreover, Jin et al. designed 3-D silk fibroin hydrogel via a bio-inspired mineralization approach of hydroxyapatite for bone tissue engineering [78]. Incorporation of calcium within the hydrogel regulated orientation growth, and the concentration of calcium modulated mineralization [78]. The compressive strength of the mineralized hydrogel correlated with the hydrogel mineral content [78].

### 6.12. Caddisfly Silk

Caddisfly silk is a strong adhesive with force stemming from post-translational phosphorylation of serine [79,80]. Inspired by aquatic caddisworm silk, the viscous natural polymer serves as an underwater cement that adheres to hard surfaces [45,80]. Caddisfly silk has the potential to be muco-adhesive due to its strong force underwater [45,80]. Stewart et al. adapted caddisfly larval silks to aquatic habitats by phosphorylating H-Fibroin serines [81]. The atmosphere of silk proteins could help assemble silk fibers via electrostatic associations of arginine-rich phosphorylated blocks [81]. The phosphorylated serine in caddisfly larval silk proteins may assist in the periodic sub-structure through calcium cross bridging [81]. Further, Lane et al. mimicked the toughening mechanism of aquatic caddisworm silk by developing a synthetic phosphate-graft-methacrylate prepolymer [80]. Viscous unfolding of ion crosslinks at critical stress that dissipates energy increases toughness [80]. The toughness of the bio-inspired hydrogel was greater than that of cartilage or meniscus [80]. In another study, Ashton et al. evaluated self-tensioning aquatic caddisfly silk [79]. FT-IR spectroscopy demonstrated that native silk has a conformation of random coils, β-sheets, and turns [79]. Replacing multivalent ions with sodium EDTA impacted fiber mechanics and made conformational changes [79]. However, the effects of EDTA could be reversed by restoring calcium [79]. Notably, molecular dynamic simulations were used to create a hypothetical structure [79]. Wang et al. identified a new bio-adhesive silk filament protein spun by Caddisfly Larvae. The protein is 98 kDa with cysteine having the largest amino acid percentage [82].

### 6.13. TAPE

Kim et al. produced a medical adhesive via inter-molecular hydrogen bonding between tannic acid and polyethylene glycol (PEG) [83]. Beyond mixing these two compounds, TAPE does not require any other chemical synthetic procedure to be formed, interestingly [83]. TAPE had substantially higher adhesion strength than fibrin glue, can be maintained in aqueous environments, and demonstrates effective hemostatic behavior [83]. This natural polymer is inspired by tannic acid, which is a degradable polyphenol compound found across almost all plant species [83]. For example, Shin et al. demonstrated that tannic acid, which is rich with pyrogallol, formulated with PEG, also known as TAPE, is muco-adhesive [84]. Keeping TAPE on esophageal mucosal layers for several hours in vivo resulted in inter-molecular interactions between the polyphenols of tannic acids and mucin that were dependent on pH and that had higher adherence in neutral conditions compared to acidic conditions [84].

### 6.14. Edible Bird’s Nest

Protein pepsin-trypsin hydrolysates found in edible bird’s nests (EBN) contain anti-oxidant peptides [85]. Ghassem et al. identified two of these pentapeptides that are natural anti-oxidants and that have the potential to serve as nutraceutical compounds [85]. Pentapeptide Pro-Phe-His-Pro-Tyr corresponds to f134–138 of cytochrome b of the swiftlet species *Aerodramus fuciphagus*, Leu-Leu-Gly-Asp-Pro to f164–168 [85]. These peptides exhibited resistance against gastrointestinal proteases, lacked cytotoxicity in vitro in human lung cells, and prevented human liver cell damage by hydroxyl radicals [85]. Further, Jessel et al. employed computational and experimental techniques to understand the structural biology of swiftlet nests on vertical rock walls with threaded saliva [86]. The team generated numerical models of the nests loaded with bird and egg forces using µCT scans to evaluate stress distribution [86]. Macro-and micro-scale structural patterns were the same across nests, indicating that the construction is managed by design principles. [86] The response to the bird and egg loads indicated a mechanical overdesign strategy that guarantees that the stress is minimized compared to the tensile fracture strength of the material [86]. This mechanical overdesign suggests a biological strategy to maintain resilient material constructions that protect the eggs [86]. Moreover, a patent by Valles-Ayoub et al. makes a claim on a nutritional supplement with acetylated neuraminate and/or a compound selected from N-acetyl mannosamine, with the invention enabling serum and cellular N-acetyl mannosamine contents to increase [87]. EBN has a high content of natural acetylated neuraminate [87]. The patent also claims regarding controlled release from muco-adhesive polymers [87].

The following section aims to highlight some of the recent and ongoing drug delivery developments.

## 7. Current Developments

### 7.1. Chitin- and Chitosan-Based Systems

Through the catechol conjugation of chitosan, Kim et al. recently demonstrated that chitosan-catechol (Chi-C) has increased muco-adhesion and efficacy as a muco-adhesive polymer with *negligible* cytotoxicity [48]. In the *contact* stage of muco-adhesion, mucin binds between Chi-C through both physical and covalent bonding; in the *consolidation* stage, catechol-mediated interaction greatly increases the duration of chitosan on the mucosal layer, thus increasing muco-adhesion [48]. The study demonstrated increased bioavailability using human insulin levels in the blood, comparing h-insulin/chitosan capsules with h-insulin/Chi-C capsules, as well as organ-dependent adhesion in mouse models, such that the primary organ for mucosal adhesion of fluorescently labeled Chi-C was the intestine, a more alkaline environment, possibly causing increased covalent interactions, whereas the primary organ for mucosal adhesion of PAA and mucin is the esophagus, which is more acidic, as previously demonstrated (sericin2) [48]. In another study, Xu et al. developed a bio-inspired enhancement of muco-adhesion of chitosan (CH) by simply mixing CH with 3 different catechols (DOPA, DA, and HCA) to create 3 different hydrogels; the presence of these catechols, particularly HCA, significantly decreases swelling and thus increases both muco-adhesion and release kinetics [46]. Furthermore, oxidation after contact with the mucosal membrane results in a supplementary increase of muco-adhesion for the HCA-CH gel, as confirmed with catechol oxidation studies and muco-adhesion tensile tests; the swelling of hydrogels was pH dependent, and HCA-CH demonstrated the least swelling and slowest catechol release, most likely because HCA carboxylic acid groups and CH amino groups interact electrostatically and reduce the electrostatic repulsion responsible for the swelling of the other hydrogels [46]. In another study, employing spray-dried microparticles with different chitosan-gelatin weight ratios, and after compression, Abruzzo et al. tested buccal tablets for adequate weight uniformity, content uniformity, friability, and hardness [88]. The greatest amount of drug permeation of propranolol hydrochloride, by-passing hepatic first-pass metabolism and improving drug bio-availability, amongst all formulations and variations of chitosan:gelatin weight were tablets with chitosan:gelatin ratios of 10:0 and 8:2, implying that an excess of chitosan demonstrates/provides the best muco-adhesive properties and best potential for increased bioavailability [88]. On the other hand, nanoparticles formed by the natural muco-adhesive polymers chitosan (CS) and dextrous sulfate (DS) using ionic gelation exhibited 5 times greater muco-adhesion to buccal cells ex vivo when compared to control CS/sodium triphosphate pentabasic (TPP) nanoparticles [89]. However, whilst the electrostatic interaction between the amine groups in the CS and the hydrogen bonding of the sulfate groups in the DS caused muco-adhesion to mucin, yet during ionic gelation, negative DS sulfate groups binded to the positive CS amine groups thereby reducing muco-adhesion of the CS/DS nanoparticles to mucin in vitro such that there is a significant difference between CS/DS nanoparticles and CS/TPP nanoparticles in vitro [89]. In another ongoing development, laminated chitosan (CS):hydroxopropyl methylcellulose (HPMC) composite sponges as a delivery matrix for tripterine (TRI)-phytosomes coated with muco-penetrating protamine, laminated with an impermeable EC layer via the electrostatic assembly process, confirmed by specific surface determination when compared with placebo sponges, was investigated as a buccal delivery mechanism to enhance bio-availability of tripterine, a potential anti-cancer *herbal* drug [90]. Briefly, the sponges loaded with PRT-TRI-PHY showed delayed drug release and superior permeation profile compared to purely TRI-PHY sponges, possibly due to the interactions between PRT and CS, resulting in superior muco-adhesion and thus higher bio-availability of PRT-TRI-PHY in in vivo pharmaco-kinetic studies in rabbits [90]. Furthermore, Garcia et al. demonstrated that 4% chitosan (CH) gels with 5% Tween 80, a penetration enhancer, have increased Toluidine Blue O (TBO) penetration compared to just 1% TBO aqueous solution, granting adequate rheological and muco-adhesive properties [91]. TBO release studies, in vivo, showed augmented TBO penetration and TBO retention at the applied site, possibly attributed to the muco-adhesive nature of CH and enhancer effect of CH and TW [91]. In another study, non-mulberry Samia cynthia (S. c.) ricini fibroin was successfully prepared into smooth, granule-free films with casting regenerated S. c. ricini solution through a step-wise decrease in urea concentration dialysis, facilitating rearrangement of fibroin [92]. On S. c ricini fibroin films, in vitro, L929 murine fibroblast cells demonstrated greater muco-adhesion, proliferation, and spreading than on B. mori (mulberry) fibroin films, possibly governed by initial non-specific binding as well as by protein layer adsorption composed efficiently for cell adhesion due to adequate hydrophobic and electrostatic amino acid interactions [92]. Last but not least, Xu et al. used hydrogels with modified catechol, chitosan, and crosslinked genipin (Cat-CS) hydrogels for buccal delivery [93]. Using 9% and 19% catechol conjugation, catechol was covalently bonded to the backbone of chitosan and cross-linked with genipin, which is non-toxic [93]. The gelation time and mechanical properties of Cat-CS hydrogels were comparable to chitosan-only hydrogels but improved mucoadhesion in vitro [93]. The Cat-CS hydrogels could detect lidocaine for about 3 h in rabbit buccal mucosa in vivo, only at 1 ng/mL, without inflammation [93]. Xu et al. also recently explored using hydrogels with modified catechol, chitosan, and crosslinked genipin (Cat-CS) hydrogels for rectal delivery of sulfasalazine in patients with ulcerative colitis [6]. Compared to oral delivery, rectal delivery with Cat-CS hydrogels had improved efficacy, histologic scores, topical therapeutic benefits, and safety, while also preventing side effects, because the drug is not delivered into small intestine [6]. At the same time, the efficacy of the rectal hydrogels was diminished, due to in-colon retention time [6].

### 7.2. Alginates and Alginate-Based Systems

Shtenberg et al. developed an oral alginate-liposome muco-adhesive delivery system with a cross-linked paste, against oral cancer [94]. The liposome was added to prevent drug degradation and enhance absorption. The hybrid cross-linked paste had 80% retention on tongue/lingual tissue compared to 50% in polymers without cross-linked paste, with a release rate of 20% after 2 h. Alginate paste also promoted cancer cell death [94]. On the other hand, Kilicarslan et al. used chitosan-alginate muco-adhesive polymers loaded with clindamycin phosphate against periodontitis [95]. Results were best when alginate was 3x higher than the concentration and volume of the polymer and when combined with low molecular weight chitosan [95]. Furthermore, Ghumman et al. designed muco-adhesive microspheres composed of *linum usitatissimum* mucilage and alginate and then loaded with metformin HCl. Microspheres had hypoglycemic effects in diabetic rats induced by alloxan hydrate [96].

**Table 2 polymers-14-05459-t002:** Bio-inspired Muco-adhesive Polymer and their Classification.

Polymer	Source	Chemical Composition	Bio-Inspiration	Elastic Modulus	References
Aggregate Silk Glue	Natural	64-mer (Gly-rich): Met-Gly-Tyr-Lys-Lys-Thr-Val-Gly-Lys-Asp-Gly-Gln-Ile-Val-Tyr-Thr-Met-Thr-Glu-Thr-Tyr-Gly-Gly-Ser-Gly-Gly-Asn-Gly-Gly-Asn-Gly-Gly-Asn-Gly-Gly-Pro-Gly-Gly-Asn-Gly-Gly-Asn-Gly-Gly-Pro-Ser-His-Gln-Thr-Pro-Gly-Gly-Gly-Ala-Pro-Gly-Met-Ser-Ser-Ser- Glu-Leu-Thr-Ala 36-mer (X_1_-Pro-Gly-X_2_-Gly, where X_1_ is Gln, Glu, or Arg and X_2_ is Ser or Asn): Gln-Pro-Gly-Asn-Gly-Gln-Pro-Gly-Ser-Gly-Gln-Pro-Gly-Ser-Gly-Glu-Pro-Gly-Ser-Gly-Gln-Pro-Gly-Ser-Gly-Gln-Pro-Gly-Tyr-Tyr-Arg-Pro-Gly-Gly-Lys-Gly 33-mer (Gly-Gly-X_1_/Asn-X_2_-Asn-X_2_-Asn, where X_1_ is Ala, Gly, Leu, or Ser and X_2_ is Val, Asp, Leu, Phe, or Met): Gly-Gly-Gln-Ser-Gly-Gly-Gly-Gly-Asn-Tyr-Asn-Val-Asn-Leu-Asn-Gly-Gly-Gly-His-Gly-Gly-His-Pro-Gly-Gly-Ser-Leu-Asn-Val-Asn-Ala-Asn-Gly	*Araneoid* orb-weaving spider silk glues	0.1–0.4 mN	Opell et al.; Brooks et al.; Petrou et al.; Elices et al.; Elices et al.; Sahni et al.; Vasanthavada et al. [45,64,97,98,99,100]
Alginate	Natural	C_12_H_20_O_12_P_2_	Alginate-thiol	~6500 mN	Davidovich-Pinhas et al.; Zia et al.; PubChem [55,56,101]
Caddisfly Silk	Natural	O-phospho-ser cluster (Ser-X)_n_; X = Val, Leu, Ile, or Arg; *n* = 2–6	Aquatic caddisworm	32.7 ± 6.6 MPa (stress at fracture)	Lane et al.; Brooks; Wang et al.; Stewart et al. [45,80,81]
Carbopol 934-P	Synthetic	(C_3_H_4_O_2_)_n_	Carbomer	0.3–13 Pa	Singla et al.; Bera et al.; Tamburic et al.; Takeuchi et al.; Blanco-Fuente et al.; NIH [58,60,61,62,102]
Chitosan + Derivatives	Natural	(C_6_H_11_NO_4_)_n_	Shellfish, insects, fungi	32.4 ± 14.5 mN; 39 to 67 Pa	Felt et al., Dash et al.; Brooks; Lehr; Kim et al.; Sogias et al.; He et al.; Cho et al.; Zvarec et al.; Ways et al.; Shitrit et al.; Snyman et al.; Elgadir et al.; Kumar et al. [42,44,45,51,52,84,103,104,105,106,107]
Mussel Adhesive Protein (MAP)	Synthetic	[Ala-Lys-Pro-Ser-Tyr-Hyp-Hyp-Thr-Dopa-Lys]_80_	DOPA, Mussel adhesive proteins from blue mussel (*Mytilus edulis*)	*Uncertain*	Ryu et al.; Schnurrer et al.; Lee et al.; Deacon et al.; Lim et al. [49,52,53,54,108]
Pectin-Sodium Carboxymethyl cellulose System	Natural	1:1:2 ratio of carbopol, pectin, and sodium carboxymethylcellulose	Pectin and sodium carboxymethyl-cellulose	23.2 ± 6.2 mN	Gupta et al. [57]
Pepsin-trypsin hydrolysates	Natural	Pentapeptides Pro-Phe-His-Pro-Tyr and Leu-Leu-Gly-Asp-Pro in f134–138 and f164–168 of cytochrome b, respectively	Swiftlet species *Aerodramus fuciphagus* of edible bird’s nest	155 MPa	Ghassem et al.; Jessel et al.; Valles-Ayoub et al. [85,86,87]
Pyriform Silk	Natural/Synthetic	Two repetitive motifs: Gln-Gln-Ser-Ser-Van-Ala and Pro-X-Pro-X-Pro, where X is a variable amino acid residue	*N. clavipes* pyriform silk	39.8 ± 8.9 mN	(Natural) Brooks; Wolff et al.; Blasingame et al.; Geurts et al. (Synthetic) Opell et al.; Brooks; Petrou et al.; Elices et al.; Peng et al. [45,64,65,66,67,68,97,109]
Sericin	Natural	[Ser-Ser-Thr-Gly-Ser-Ser-Ser-Asn-Thr-Asp-Ser-Asn-Ser-Asn-Ser-Val-Gly-Ser-Ser-Thr-Ser-Gly-Gly-Ser-Ser-Thr-Tyr-Gly-Tyr-Ser-Ser-Asn-Ser-Arg-Asp-Gly-Ser-Val]_n_	Silkworm-derived adhesive	4.1 ± 2 N	Jiang et al.; Brooks; Freitas et al. [45,74,110]
Silkworm fibroin	Natural	Gly-Ala-Gly-Ala-Ser, Gly-X_n_; X = Ala, Tyr, or Val	Silkworm-derived adhesive	54 mN or 1466 Pa	Brooks; Jiang et al.; Yucel et al.; Serban et al.; Kundu et al.; Wei et al. [45,69,71,78,110,111]
TAPE	Natural	1 g mL^−1^ in distilled water of tannic acid (C_76_H_52_O_46_) blended with 1 g mL^−1^ in distilled water of PEG (C_2n_H_4n+2_O_n+1_)	Tannic acid in plants	Up to 1 kPa	Kim et al.; Shin et al. [83,84]

## 8. Conclusions and Future Perspectives

Overall, muco-adhesion is a very attractive pharmaceutical and user-friendly strategy to prolong the residency time as well as the contact between tissues/membranes and bio-active drug or pharmaceutical formulations. Thereby, such facilitates a sustained and controlled drug delivery and hence, reduces the frequency of need and administration, for the bio-effect. Indeed, the mechanisms, processes, theories and properties of muco-adhesion and, more particularly, the muco-adhesive bio-polymers, have gained increasing interest in the last couple of decades, leading to novel solutions, innovative devices and even commercialized products, and by default, enhanced investments in R&D&I (Research, Development and Innovation). Overall, muco-adhesive drug delivery systems demonstrate great potential in improving the bio-availability and increasing the therapeutic effects of drugs and the encapsulated bio-active load. The buccal (*trans-buccal*) delivery route, in specific, has been the most commonly investigated, thus far, to maximize drug deposition and retention time. Recent advances focus on further improving some of the limitations, including, permeability, stability at the residence or adhesion site, drug loading enhancement, and muco-adhesion strength increase. Furthermore, it is noteworthy that drug delivery strategies, herein, are trending more towards the use or incorporation of nanoparticles and/or malleable hydrogels and/or multi-compartmental thin-films, for better pharmaco-kinetics and -dynamics [3,111,112,113,114]. Researchers seem to be also turning attention or preference towards the next-generation of biocompatible polymers, bio-inspired and bio-mimetic polymers [3,115], that besides their muco-adhesive properties, may possess or provide novel/desired features that may further enhance the drug retention rate/bio-availability, controlled biodegradability, predictable performance and behavior in situ, or even the drug loading capacity and encapsulation efficacy, as such parameters would greatly impact the pursued release kinetic profiles and dose–response curves [112,113,114]. Furthermore, some potential muco-adhesive systems are being explored with different synthetic rate-controlling agents such as poly (acrylic acid)-based polymers. Polymer manipulation is essential for controlling the absorption rate and bio-availability of drugs, which may lead to promote advanced treatment approaches and superior therapeutic outcomes.

## Figures and Tables

**Figure 1 polymers-14-05459-f001:**
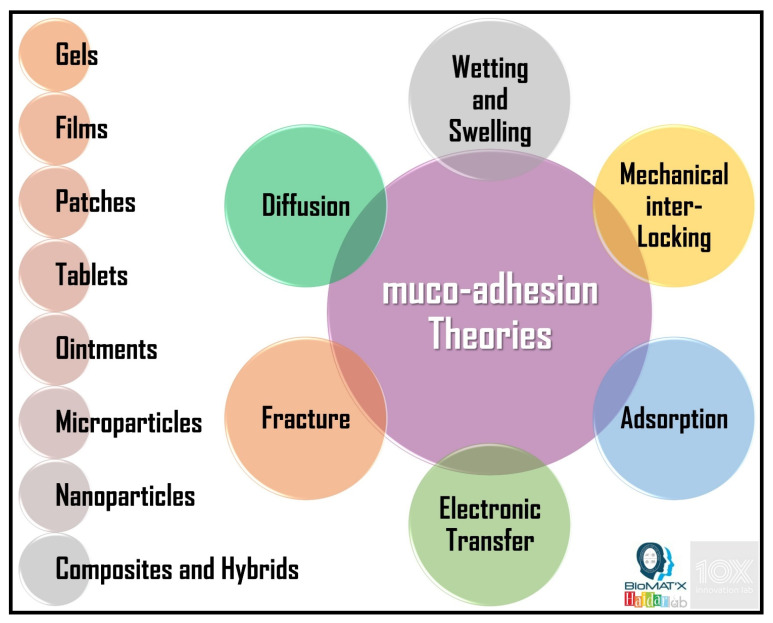
Theories of muco-adhesion and possible types or formats of muco-adhesive formats.

**Figure 2 polymers-14-05459-f002:**
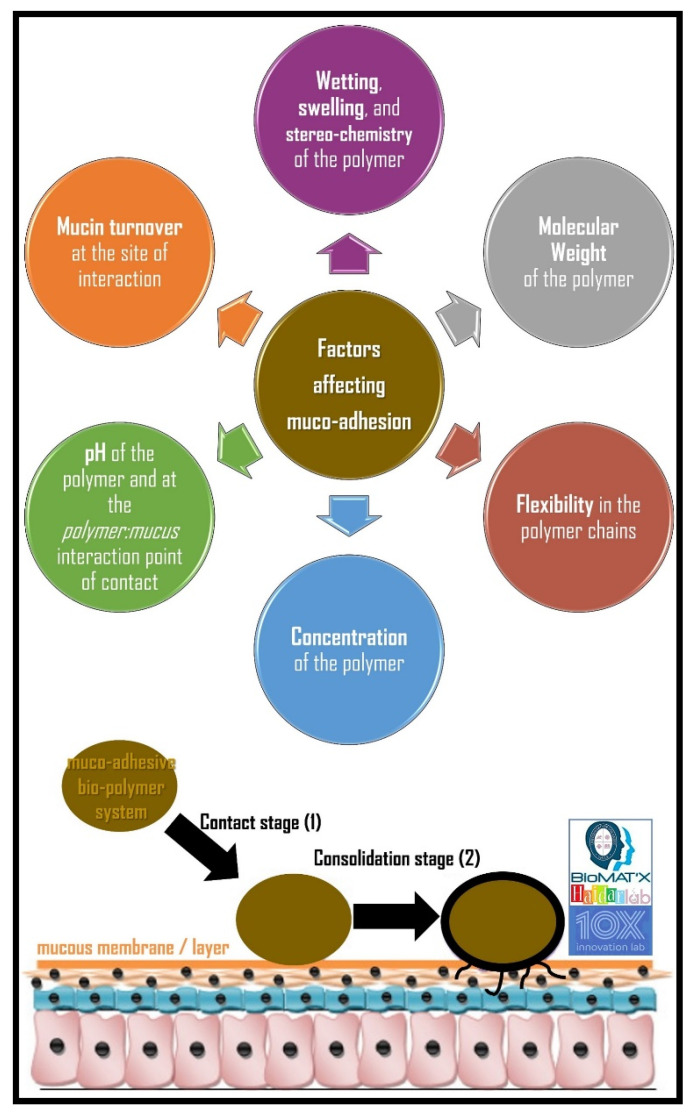
Two-stage mechanism and factors involved in muco-adhesion and drug delivery.

**Figure 3 polymers-14-05459-f003:**
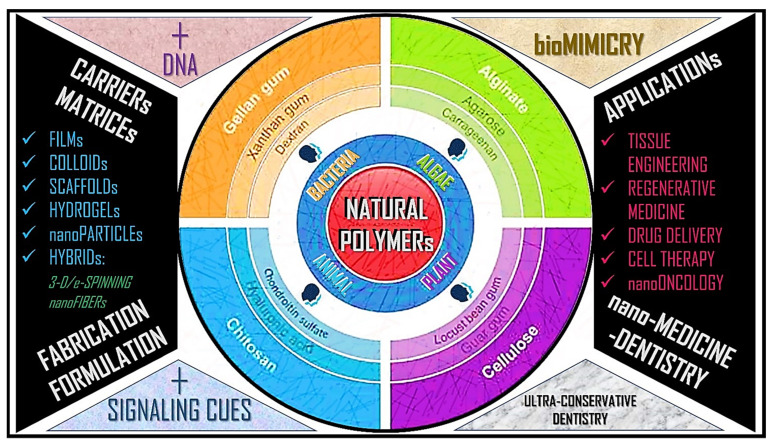
Bio-inspired/-mimetics polymers for muco-adhesive drug delivery, and beyond.

**Table 1 polymers-14-05459-t001:** Summary of the Major Muco-Adhesion Theories.

Theory	Summary	Equation(s)	Diagram	Label (s)
Wetting	The wetting theory is applicable to liquid bio-adhesives. It treats adhesion as an embedding process, whereby the lower the contact angle, the greater the adhesion. In this process, adhesive agents penetrate surface irregularities to spread through the surface.	γ*_TA_* = γ*_PT_* + γ*_PA_*cos(*θ*)	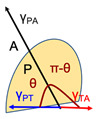	*T*: Tissue; substrate surface; mucosal membrane *P*: Polymer; liquid; mucoadhesive material *A*: Air; vapor *θ*: Angle of contact angle between solid and liquid interface γ*_TA_*: Interfacial tension between tissue and air γ*_PT_*: Interfacial tension between polymer and tissue γ*_PA_*: Interfacial tension between polymer and air
Mechanical Inter-locking or Keying	Mechanical inter-locking theory, or mechanical keying, proposes that the adhesion between a liquid and a rough or porous surface is due to the mechanical inter-locking as well as the increased surface rugosity.	OJS = C ∗ MIC ∗ ICC	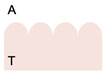	OJS: Optimum joint strength C: Constant MIC: Mechanical interlocking component ICC: Interfacial chemical component *A*: Air *T*: Tissue
Adsorption	The adsorption theory states that adhesion is caused by molecular bonding between the mucus membrane and muco-adhesive device.	N/A	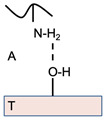	*T*: Tissue *A*: Air
Electronic	The electronic transfer theory proposes that electron transfer between the mucus membrane and the muco-adhesive substrate results in attractive layers within a double layer of electrical charges, at the interface.	N/A	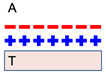	*T*: Tissue *A*: Air
Fracture	A commonly used theory in mechanical measuring muco-adhesion, the fracture theory, calculates the force necessary to detach two surfaces after adhesion is established.	*S_m_* = *F_m_*/*A*_0_*σ* = ((*E* ∗ *ε*)/*L*)0.5*ε* = *W_r_* + *W_i_E* = [*σ*/*ε*]*ε*→ 0 = [[*F*/*A*_0_]/[∆*l*/*l*_0_]]_∆*l*→0_	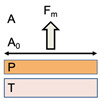	*S_m_*: Adhesive strength/detachment force *F_m_*: Maximal force for detachment *A*_0_: Surface area encompassed by muco-adhesive system *T*: Tissue *A*: Air *P*: Polymer muco-adhesive *σ*: Fracture strength *E*: Young’s modulus for elasticity *ε*: Fracture energy *L*: Critical crack length *W_r_*: Reversible adhesion work *W_i_*: Irreversible adhesion work ∆*l*: Change in thickness *l*_0_: original thickness
Diffusion	A semi-permanent adhesive bond is formed via the deep inter-penetration and entanglement of the polymer and mucin chains, with the adhesive force being proportional to the degree of penetration.	l = (t ∗ Db)^½^	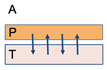	I: Interpenetration depth t: Contact Time Db: Diffusion co-efficient of mucoadhesive *T*: Tissue *P*: Polymer muco-adhesive *A*: Air

## Data Availability

Not applicable.

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
