# Peer review of "Bio-Inspired Muco-Adhesive Polymers for Drug Delivery Applications"

_polymers, 2022, doi:10.3390/polym14245459_

Round 1
Author Response
The authors thank the reviewers for the provided corrections and suggestions.
Summary of performed corrections:
Reviewer 1
- Caption for Figure 1 (and the others) is now edited, as suggested by Reviewer 2. The sentence "Figure 1 below illustrates the main ..." mentions "below" to indicate the positioning (placement location) of the figure in the final manuscript.
- Figure 1 is provided at end of manuscript, for the journal's use in formatting. All figures have been re-placed into the text as well, as requested.
- Table 1 title is "Summary of the Major Muco-Adhesion Theories". Its mention in the manuscript text has been edited, and "electronic" added.
- Comments/suggestions 4-6 have been checked to the best of ability, and mainly are formatting-related, hence, some have been left to the final copy-editing/proof-reading stage.
- Formula 6 verified.
- Section 4 is now merged into Section 6, and Sections re-named to highlight the content relevance.
- Conclusions revised, and we thanks the reviewer for his kind comment. Indeed, it is in an ongoing manuscript by one of our thesis students.
Reviewer 2
- Language improved throughout the manuscript, and lengthy sentences edited. Most changes have been highlighted in yellow.
- Section 4 combined with Section 6. Numbering adjusted.
- Examples of drug delivery applications incorporated into the relevant sections, as suggested.
- Formatting of the manuscript improved. All detected capitalizations and other formatting typos/errors are now corrected.
- ‘electronic’ added to the caption of Table 1.
- Figure 3 clarity adjusted to the best of ability.
- Dopamine is now mentioned once in the mentioned sentence.
Reviewer 3
- Besides the preformed editing mentioned above, the first 10 lines of the introduction have been revised, including referencing.
- Examples of muco-adhesive polymers are extensively described and discussed in the manuscript as well as in the illustrations.
- Section 2 is deemed necessary to explain to the inexperienced and interested reader, via this comprehensive review, the relevant muco-adhesion theories, to simplify their understanding, and application. We hope the reviewer can accept this Section to remain, as the other 2 reviewers. Thank you.
- Route comparison has been mentioned within the text (Section 4), through the applied examples provided. Perhaps solely can be the topic/focus of a separate and dedicated manuscript. Interesting idea.
- Sabbagh, F., & Kim, B. S. (2022). Recent advances in polymeric transdermal drug delivery systems. Journal of Controlled Release, 341, 132-146, has been cited. now it is Reference #3.

Reviewer 2 Report
This review describes an overview of bio-inspired muco-adhesive polymers for drug delivery applications. I found this review clearly structured and very informative. It covers this field comprehensively as much as possible. Overall, the paper is well written and provides an exhaustive update of the field. Accordingly, it is worthy of publication after a minor revision, and some comments are listed in the following.
General Comments:
1. Although the overall structure of this manuscript is organized, language needs to be improved to enhance readability. Specifically, there are many lengthy sentences, especially in the introduction part (e.g. line 80-87), which make it hard to deliver clear and concise information.
2. Section 4 (Characteristics of bio-polymers) is more relevant to Section 6 (Muco-adhesive bio-polymers). I would suggest that the authors change the order of sections or combine Section 4 with Section 6.
3. The topic of this manuscript is about bio-inspired muco-adhesive polymers for drug delivery applications, while the authors focused more on discussing the materials. Could the authors provide more examples of using those materials for drug delivery applications in Section 7 and Section 8?
Detailed Comments:
1. Formatting of the manuscript needs to be improved. For example, capitalization of the first word in the sentence (e.g. line 53, 65, 80, 108, 310, 318, 328, 425, 435, 563), an extra word (line 245), an extra spacing (line 39, 103, 137, 426).
2. Line 231, ‘electronic’ is missing in the caption of Table 1.
3. Figure 3 is a little bit blurred. Please use a high-quality figure instead.
4. Line 512-514, dopamine was mentioned twice in this sentence.
Author Response

(The authors gave the same response as above.)

Reviewer 3 Report
1- Try not to use bulk data from one reference. There are no references in the first 10 lines of the introduction.
2- What are examples of mucoadhesive polymers? Which one is better in release studies? Compare the different past related studies in a Table.
3- Is section 2 necessary to discuss in a review paper? Some experiments are explained coming with the related formula. This section can remove totally.
4- It is a good idea if the authors prepare a Table and compare the obtained data from different routes of drug delivery using mucoadhesive polymers.
5- The authors can use the following reference in this manuscript to improve some parts such as the introduction or discussions:
Sabbagh, F., & Kim, B. S. (2022). Recent advances in polymeric transdermal drug delivery systems. Journal of Controlled Release, 341, 132-146.
Author Response

(The authors gave the same response as above.)
